# Genetic Abnormalities in Multiple Myeloma: Prognostic and Therapeutic Implications

**DOI:** 10.3390/cells10020336

**Published:** 2021-02-05

**Authors:** Ignacio J. Cardona-Benavides, Cristina de Ramón, Norma C. Gutiérrez

**Affiliations:** 1Hematology Department, University Hospital, Institute of Biomedical Research of Salamanca (IBSAL), University Hospital of Salamanca, 37007 Salamanca, Spain; icarbe96@usal.es (I.J.C.-B.); cramon@usal.es (C.d.R.); 2Cancer Research Center-IBMCC (USAL-CSIC), 37007 Salamanca, Spain; 3Centro de Investigación Biomédica en Red de Cáncer (CIBERONC), Spain

**Keywords:** chromosomal translocations, copy number changes, point mutations, multiple myeloma, prognosis, targeted therapy

## Abstract

Some genetic abnormalities of multiple myeloma (MM) detected more than two decades ago remain major prognostic factors. In recent years, the introduction of cutting-edge genomic methodologies has enabled the extensive deciphering of genomic events in MM. Although none of the alterations newly discovered have significantly improved the stratification of the outcome of patients with MM, some of them, point mutations in particular, are promising targets for the development of personalized medicine. This review summarizes the main genetic abnormalities described in MM together with their prognostic impact, and the therapeutic approaches potentially aimed at abrogating the undesirable pathogenic effect of each alteration.

## 1. Introduction

Multiple myeloma (MM) is a genetically complex and heterogeneous neoplasm in which the concurrency of multiple genomic events leads to tumor development and progression (Figure 1). Moreover, genetic abnormalities are major prognostic factors in MM. Molecular cytogenetic methodologies such as G-band karyotyping, fluorescence in situ hybridization (FISH), and comparative genomic hybridization (CGH) combined with more advanced genetic techniques, encompassing single nucleotide polymorphism (SNP) arrays and, more recently, next-generation sequencing (NGS) [1], have made it possible to identify numerous recurrent chromosomal and genetic alterations in MM that can be categorized into three types: chromosomal translocations, copy number abnormalities (CNAs), and point mutations [2]. Many of these abnormalities have been tested in several powerful studies involving a large number of patients. The new revised international staging system (R-ISS), developed by the International Myeloma Working Group (IMWG), requires the analysis of translocations t(4;14) and t(14;16), and 17p deletion to stratify risk in MM patients [3].

Most of the abnormalities present in MM patients are not druggable, but progress in biological, clinical, and therapeutic research has helped identify drugs and combinations thereof that are more effective for treating particular abnormalities, as well as making several specific target agents available.

## 2. Genetic Abnormalities

### 2.1. Chromosomal Translocations

The most common translocations observed in MM patients are those involving the *IGH* gene (14q32), which is translocated to diverse oncogenes whose expression is upregulated under the influence of the powerful *IGH* enhancer. Higher levels of expression of certain oncogenes may confer a selective advantage on those subclones that carry the translocations [4]. *IGH* translocations are considered initiating events and are therefore named primary translocations. They are present in up to 50% of patients, and mainly involve five chromosomal loci, 11q13, 6p21, 4p16, 16q23, and 20q11, that contain the *CCND1*, *CCND3*, *FGFR3/NSD2*, *MAF*, and *MAFB* oncogenes, respectively. These translocations lead to the overexpression of the oncogene juxtaposed to the 3’ intronic *IGH* enhancer, and, in the particular case of t(4;14), provoke the simultaneous dysregulation of two oncogenes [5].

Rearrangements of the *MYC* oncogene, which cause it to be overexpressed, occur frequently in MM, and are considered secondary cytogenetic events. *MYC* translocations are observed in approximately 15–20% of patients newly diagnosed with MM [1,6].

#### 2.1.1. Translocations t(11;14) and t(6;14)

Translocations t(11;14) and t(6;14) juxtapose the *IGH* enhancer with *CCND1* (15–20%) and *CCND3* (1–4%), respectively. Most of them occur at the switch region breakpoints, although a subset of *CCND1* translocations has been described as originating from errors in V(D)J recombination [7]. The cyclin D dysregulation induced by both translocations inactivates RB1 (retinoblastoma), allowing cell-cycle progression [8].

##### Prognosis

Traditionally, newly diagnosed MM (NDMM) patients with t(11;14) have been categorized as standard risk [9,10] (Table 1). However, some recent studies indicate that t(11;14) is associated with shorter survival in the era of novel agents, acting as a marker of intermediate risk [11,12,13,14]. These data also suggest that high-dose chemotherapy benefits patients with t(11;14) [13,15]. Although the low prevalence of t(6;14) precludes the estimation of survival, this translocation is included in the standard-risk category [10]. Patients with one of these translocations are more likely to have bone disease [16].

##### Therapeutic Implications

In vitro experiments with MM cell lines and primary samples with t(11;14) treated with venetoclax revealed a high degree of sensitivity to this BCL2 inhibitor. These preclinical results were confirmed when patients with relapsed or refractory multiple myeloma (RRMM) bearing t(11;14) were treated with venetoclax in monotherapy, yielding an overall response rate (ORR) and median progression-free survival (PFS) of 86% and 6.6 months, respectively. Responses to venetoclax were associated with higher ratios of BCL2/BCL2L1 or BCL2/MCL1 [1,17]. One of the mechanisms of resistance to venetoclax is the overexpression of Mcl-1. Bortezomib has been reported to overcome this resistance by upregulating Noxa and subsequently neutralizing Mcl-1 [18]. On this basis, venetoclax has been tested in combination with dexamethasone and bortezomib in a phase Ib study, which yielded promising results in terms of response (ORR 67%) and survival (median PFS 9.5 months) [19]. Recently, a phase 3 trial testing the addition of venetoclax to bortezomib and dexamethasone showed significantly increased response rates (ORR 85% vs. 70%) and PFS (not reached vs. 9.9 months, *p* < 0.0001) without increasing mortality in patients with high BCL2 levels or t(11;14). However, the mortality in the experimental arm was higher in patients without t(11;14) due to increased infection rate [20,21]. According to these data, two phase 2 trials with venetoclax in combination with carfilzomib, and with daratumumab plus bortezomib, are now recruiting, and a phase 3 trial involving a combination with pomalidomide and dexamethasone is progress. Several clinical trials with Mcl-1 inhibitors, such as ABC294640 (sphingosine kinase 2 inhibitor), and with AZD5991 are underway [22,23,24,25]. AZD5991 is a small Mcl-1 inhibitor that activates the Bak-dependent mitochondrial apoptotic pathway [24], and its combination with venetoclax has shown synergism. Furthermore, dexamethasone enhances the expression of Bcl-2, which sensitizes MM cells to these kinds of agents [25].

#### 2.1.2. Translocation t(4;14)

This translocation is observed in approximately 15% of patients, its incidence decreasing with age [5,26]. Translocation t(4;14) leads to simultaneous overexpression of two genes, *FGFR3* (fibroblast growth factor receptor 3) and *NSD2* (nuclear receptor binding SET domain protein 2), the latter also known as *MMSET* (multiple myeloma SET domain protein) and *WHSC1* (Wolf–Hirschhorn syndrome candidate 1) [8]. Whereas *MMSET* is overexpressed in all cases with t(4;14), overexpression of *FGFR3* is observed in 70% of patients with t(4;14) due to the deletion of the der(14) chromosome in one-third of cases. This suggests that *MMSET* activation may play a more critical role than *FGFR3* in the pathogenesis of MM with t(4;14), contributing to increased proliferation, a change in cellular adhesion, and high tumorigenicity [27]. *MMSET* encodes histone 3 lysine 36 (H3K36) methyltransferase, and its overactivation in MM has been shown to affect the expression of several genes, including some related to cancer [8,28]. High levels of expression of *CCND2* have been observed in myeloma cells with t(4;14), although the underlying mechanisms are not completely understood [29,30]. *FGFR3* encodes a receptor tyrosine kinase for fibroblast growth factors and plays an essential role in regulating cell proliferation, differentiation, and apoptosis. The cytotoxic effect of targeting *FGFR3* in MM cells with t(4;14) is also evidence of the contribution of *FGFR3* in myelomagenesis.

##### Prognosis

The t(4;14) translocation was associated with unfavorable prognosis in several large studies [5] and patients bearing it were classified in the high-risk category [9,31,32] (Table 1). Several studies demonstrated that bortezomib overcomes the negative prognostic impact of t(4;14) (Table 2). Induction therapy based on bortezomib-dexamethasone along with high-dose melphalan improved event-free survival and overall survival (OS) of NDMM patients with t(4;14) [33]. Bortezomib in combination with thalidomide and dexamethasone (VTD) followed by double autologous hematopoietic stem-cell transplantation (ASCT) could overcome the bad prognosis associated with t(4;14) [34,35], though this drug applied on maintenance hardly had any influence on prognosis compared with thalidomide maintenance [36,37].

In RRMM patients with t(4;14), the combination of lenalidomide and dexamethasone has been reported to achieve an OS similar to patients without this cytogenetic abnormality, suggesting that this regimen could also overcome its poor prognosis [38]. On the other hand, a retrospective study showed lower ORRs and shorter median PFS and OS in RRMM patients with t(4;14) using the same scheme [39]. The difference between the two studies could be related to differences in age, in the median number of previous therapies, and in the proportion of patients with earlier thalidomide or bortezomib therapy [39]. Strikingly, pomalidomide, despite also being an immunomodulatory drug (IMID), did not seem to benefit this group of patients [40]. Combining elotuzumab with lenalidomide and dexamethasone produced interesting results such as a PFS of about 20 months for patients with t(4;14), similar to the survival observed in standard-risk patients [41]. Moreover, the proteasome inhibitor (PI) carfilzomib used as monotherapy was able to improve the ORR and OS by up to 12 months in patients with RRMM and t(4;14) [42] (Table 2).

##### Therapeutic Implications

Targeted therapy has been employed to treat patients with MM bearing t(4;14) that uses monoclonal antibodies (mAbs) that block ligand binding sites to FGFR3. Most of these mAbs are designed to recognize the mutant residue serine249 (S249) of FGFR3 since it results in ligand-independent activation of FGFR3. Mouse studies using R3Mab and MFGR1877A mAbs revealed that both of them induce antibody-dependent cell-mediated cytotoxicity [43,44]. Additionally, PRO-001 mAb was demonstrated to bind to FGFR3 and inhibit FGFR3 autophosphorylation and downstream signaling [45]. Moreover, the activity of dovitinib, a small molecule inhibitor of FGFR3, was tested in vitro and in xenograft animal models carrying t(4;14) [46]. However, phase II trials were only able to demonstrate the stabilization of the disease [47,48]. Further studies with FGFR3 mAbs in combination with other treatments, such as chemotherapy, bortezomib, or lenalidominde, are needed to optimize anti-FGFR3 antibody-based treatment. MMSET is involved in pathways related to DNA repair, modulating the response to chemotherapy [49,50]. However, at present, inhibitory molecules against this gene are, at best, at the developmental stage.

#### 2.1.3. Translocations t(14;16) and t(14;20)

Translocations t(14;16), observed in 5% of MM, and t(14;20), seen in fewer than 2% of patients, deregulate the *MAF* and *MAFB* genes, respectively. Both genes belong to the MAF family and are leucine zipper-containing transcription factors. Increased levels of *MAF* induce upregulation of cyclin D2 through its transactivation function, resulting in an accelerated rate of division and DNA synthesis [8,29,51]. Overexpression of *MAFB* in MM induces proliferation and protects cells from drug-induced apoptosis, conferring resistance. Myeloma cells with t(14;16) or t(14;20) present genetic instability, similar to that observed for t(4;14) [52].

##### Prognosis

Translocations t(14;16) and t(14;20) are considered high-risk cytogenetic factors [53] (Table 1). The *MAF* molecular subgroup has also been associated with high levels of free light chain in serum and therefore a higher incidence of acute renal failure [54]. Although data from the Mayo Clinic associated t(14;16) with poor prognosis [55], these results have not been corroborated by the IFM group [56]. Nevertheless, this discrepancy could be related to the fact that the patients from the first study were treated with conventional chemotherapy, whereas more than 50% of the patients from the second study received a double transplant.

##### Therapeutic Implications

*MAF* transcription may be regulated by the MEK–ERK pathway. Thus, the extracellular signal-regulated kinase (ERK) has been demonstrated to activate the transcription factor FOS (a subunit of AP1), which is bound to the *MAF* promoter, inducing its transcription. Some studies reported that MEK inhibition induces apoptosis of MAF-expressing myelomas and blocks survival signals provided by the microenvironment [57,58,59]. The blockade of FOS activity is also toxic for MM cells harboring t(4;14), which is related to the *MAF* upregulation detected in myeloma cells with MMSET translocation.

AZD6244 is a novel oral and highly specific MEK inhibitor that targets not only MM cells, but also the microenvironment. Its combination with the HSP90 inhibitor TAS-116 is known to enhance the anti-myeloma effect [59]. Trametinib (GSK1120212) is a reversible MEK inhibitor that could improve ORRs when combined with AKT inhibitors [60]. Some studies show that the presence of t(14;20) in MM cell lines confers resistance to PIs, such as bortezomib [61,62]. Bortezomib leads to the stabilization and accumulation of phosphorylated MAF, which could partially explain the low efficacy of this therapy. Consequently, the inhibition of the serine/threonine protein kinase GSK3, which is responsible for the MAF and MAFB phosphorylation, could block proliferation and colony formation of MAF-expressing cells [63].

#### 2.1.4. Translocation of MYC Oncogene

Most of the *MYC* rearrangements bring about the juxtaposition of a super-enhancer adjacent to *MYC* [64], which leads to increased expression levels of *MYC* mRNA [65]. *MYC* overexpression results in an increased DNA replication rate. This causes DNA damage and an increase in reactive oxygen species (ROS). *MYC* lesions are also one of the most important events promoting disease progression [66,67,68] and have recently been associated with a high tumoral burden [69].

##### Prognosis

Over time, *MYC* deregulation has come to be regarded as a central event of MM biology. A MYC activation signature was identified in the transition from premalignant conditions of plasma cell dyscrasias to symptomatic myeloma [66,70,71]. The impact of *MYC* rearrangements in the prognosis of MM patients is controversial. A large study by the IFM group did not find any association between *MYC* translocations assessed by FISH and MM prognosis [72]. Conversely, other groups had inferior survival in patients with *MYC* rearrangements compared to patients without these abnormalities [5,6,73,74]. Recently, another study highlighted the crucial role of *MYC* rearrangements as an independent prognostic factor in NDMM [69].

##### Therapeutic Implications

The direct inhibition of transcription factors remains a great therapeutic challenge. Different indirect ways of reducing MYC activity have been explored in MM. Thus, the amplification of *MYC* generates high levels of replicative stress and ROS, with the consequence that myeloma cells harboring this translocation could be attacked with combinations of agents that block the stress response. ATR inhibitors combined with drugs that increase ROS or bortezomib can give rise to synergistic cytotoxicity [75]. Bromodomain and extra-terminal (BET) protein belong to a protein family that interacts with acetylated histones. These interactions allow the recruitment of chromatin regulators to precise chromatin sites, which results in the expression of specific genes. Different in vitro and in vivo preclinical studies using BET inhibitors demonstrated the downregulation of *MYC* expression [76,77]. Based on those preclinical studies, the BET inhibitor CPI-0610 progressed to a phase I clinical trial. Primary targets of CPI-0610 are MYC, IKZF1, and IRF4 [78]. Recently, the therapeutic potential of the proteolysis targeting chimera (PROTAC) molecule ARV-825 was proven in MM. This molecule induced degradation of BET proteins and, subsequently, downregulation of *MYC* expression [79].

Additionally, leflunomide, an agent approved for the treatment of the rheumatoid arthritis, also exhibits anti-myeloma activity through the downregulation of MYC protein in preclinical studies [80]. Leflunomide blocks the PIM family, which stabilizes MYC, and on the other hand, its active metabolite teriflunomide induces the proteasome degradation of MYC. Moreover, leflunomide in combination with lenalidomide inhibits cell growth [80]. A phase 2 trial with leflunomide in combination with pomalidomide and dexamethasone is currently underway.

Recently, MYC has been associated with higher levels of PARP-1, which has been implicated in DNA repair and indicates poor prognosis. Accordingly, PARP inhibitors could be a treatment option for these patients, although clinical trials are needed to confirm this effect [81].

### 2.2. Copy Number Abnormalities

#### 2.2.1. Hyperdiploid/Hypodiploid

Most MM cases are aneuploid, in which there are frequent gains and losses of complete chromosomes or chromosome arms. According to the ploidy status, MM is usually categorized in hyperdiploid and non-hyperdiploid MM. The hyperdiploid (H-MM) group, which accounts for 50–60% of all MM cases, is characterized by the presence of trisomies that typically affect the odd chromosomes [82]. Hyperdiploidy seems to be an early event in MM evolution since it has been described in monoclonal gammopathy of undetermined significance (MGUS) [83]. The non-hyperdiploid MM (NH-MM) group includes hypodiploid (up to 44/45 chromosomes), pseudodiploid (44/45 to 46/47), and near-tetraploid (more than 74) cases. NH-MM is frequently characterized by the loss of chromosomes 13, 14, 16, and 22. Hyperhaploid karyotypes, as a result of the loss of nearly a haploid set of chromosomes, have also been found in MM. This MM group is monosomic for many autosomes, excluding chromosomes 3, 5, 7, 9, 11, 15, 18, 19, and 21 found in disomy. The most frequent monosomies/deletions present in hyperhaploid cases affect 17p, 1p, 13q, and 16q [84,85]. In particular, monosomy 17 or del(17p) was identified in all hyperhaploid patients.

The high proportion of chromosomal imbalances observed in MM is a clear sign of underlying genomic instability. In this context, the phenomenon of chromothripsis, defined by tens to hundreds of random chromosomal rearrangements involving localized genomic regions, has been observed in a very low percentage of NDMMs [86].

##### Prognosis

Several studies have shown that hyperdiploid patients have better response rates to treatment and longer survival than patients with other aneuploidies [87]. This favorable prognosis was observed irrespective of the therapeutic context [88,89,90]. However, not all trisomies have the same impact on survival; in particular, trisomies of chromosomes 3 and 5 improve the prognosis in patients treated with chemotherapy or bortezomib followed by ASCT, and also in those treated with non-intensive protocols, even in patients with t(4;14) [91].

On the other hand, patients treated with chemotherapy belonging to the non-hyperdiploid group are characterized by more aggressive clinical features [92] and worse outcomes [88,93], especially those with hypodiploid karyotypes. The introduction of new drugs has not significantly changed this scenario. Hyperhaploid karyotypes are associated with an adverse prognosis, even worse than that of the hypodyploid group, with an estimated 20–25% survival after five years, despite intensive treatments based on PIs, IMIDs, and tandem ASCT [85].

Finally, the presence of chromothripsis seems to confer a higher risk of relapse and shorter survival [86,94].

##### Therapeutic Implications

The search for therapeutic targets associated with gains and losses of genetic material is difficult because unlike translocations and mutations, chromosomal imbalances can influence the expression of thousands of genes with very different and sometimes opposite functions. Nevertheless, some specific CNAs involve oncogenes or tumor suppressor genes of known or potential relevance in myeloma pathogenesis, as can be seen in the following sections.

#### 2.2.2. Deletion of 1p

Deletions of 1p, del(1p), have been observed in up to 30% of patients with MM [92,95,96,97]. This incidence increases up to 60% in plasma cell leukemia [98], indicating that del(1p) may be related to clonal evolution [99]. The minimally deleted regions (MDRs) identified in the interstitial deletions are the 1p12, 1p21, 1p22.1, and 1p32.3 regions, were the *FAM46C, CDC14A, MTF2*, and *CDKN2C* genes are located, respectively [95,99,100]. The most frequent MDR is 1p22, which accounts for 15–22% of cases [95,101].

##### Prognosis

Del(1p) is associated with shorter survival in transplant-eligible patients, according to data from both the Myeloma IX trial and the IFM group [95,101]. Patients with a loss of the *CDC14A* gene, located at 1p21, also had a poor prognosis after chemotherapy treatment followed by ASCT consolidation [102]. This adverse prognosis was also confirmed in patients with RRMM treated with lenalidomide and dexamethasone [103]. *FAM46C* downregulation as a consequence of del(1p12), mutation, or both, has been reported to confer resistance to dexamethasone and lenalidomide in vitro [104], although only the *FAM46C* loss, but not the mutation, was associated with a worse outcome in patients treated with triplet induction therapies [105].

##### Therapeutic Implications

Loss of *FAM46C* is associated in MM cell lines with an increase of cell migration mediated by the activation of the PI3K/Rac1 pathway, so it is reasonable to speculate that patients with del(1p12) or *FAM46C* mutations could benefit from PI3K and Rac1 inhibitors [106].

#### 2.2.3. Gain of 1q

The long arm of chromosome 1 is gained (three copies) or amplified (more than three copies) in nearly 50% of patients with NDMM [107,108,109]. These percentages increase as the disease progresses [110,111,112], reaching up to 68% in RRMM [107]. Overexpression of the *CKS1B* gene, located at 1q21, was initially linked to myeloma cell growth and survival, and consequently to drug resistance [113,114,115]. Other genes mapped at 1q, such as *MUC1, MCL1, ANP32E*, *BCL9*, *PSMD4*, and *PDZK1,* have been proposed as candidate participants in myelomagenesis [92,116,117,118,119,120].

##### Prognosis

1q gains have been associated with poor prognosis in patients treated with chemotherapy [108,113], PIs [107], and IMIDs [121], even in transplant-eligible patients [108,113,122,123]. Lenalidomide maintenance was able to improve PFS without any impact on OS in this set of patients [124] (Table 2).

The triple combination VRD can overcome the negative prognostic impact of 1q gains occurring in isolation, but not when they are in combination with other high-risk genetic abnormalities or in the case of 1q amplification [121]. The concept of “double-hit” myeloma includes those patients with 1q amplification together with international staging system ISS stage 3, which are considered to have an ultra-high risk disease in spite of novel therapeutic regimens [105,125].

##### Therapeutic Implications

Anti-CD46 antibody conjugated with monomethyl auristatin F was tested in preclinical studies with successful outcomes. This drug is especially appealing in patients with gain(1q), given that the *CD46* gene is located on the long arm of chromosome 1 and the CD46 antigen is overexpressed in this group of patients [126]. In this regard, in vitro studies recently demonstrated higher Mcl1 expression and greater efficacy of Mcl1 inhibitors for MM samples harboring amp1q [127], although clinical trials are required to confirm these results. A phase 1 clinical trial is currently in progress.

#### 2.2.4. Deletion of 13q

Deletion of 13q, del(13q), detected by FISH, is present in around 45% of patients, taking into account both monosomy 13, which is the most frequent alteration involving chromosome 13, and interstitial deletions observed in up to 15% of cases [128,129]. Its actual incidence was ascertained through the introduction of FISH with locus-specific probes [130,131]. The tumor suppressor *RB1* is lost in these chromosome 13 aberrations.

##### Prognosis

Del(13q) was described in MM for the first time in 1995 and has consistently been related to adverse prognosis [132] in almost all large series of patients treated with both conventional and high-dose therapy [93,130,133]. However, increasing evidence suggests that this unfavorable prognosis arises from its close association with other high-risk genetic features, such as t(4;14), which appears in combination with del(13q) in 80% of cases. Therefore, *RB1* loss on its own would not be a negative prognostic factor [134]. An intriguing opposite effect of monosomy 13 and del(13q) was recently described in NDMM patients treated with PIs and/or IMIDs, so del(13q) was shown to have an independent favorable impact on OS, whereas monosomy 13 was associated with shorter OS, regardless of the co-occurrence of high-risk cytogenetic abnormalities [135].

##### Therapeutic Implications

*RB* deletion seems to increase PDL1 expression, mediated by the NFkB pathway [136], so PDL1 inhibitors could be more active in this subset of patients.

#### 2.2.5. Deletion of 17p

The prevalence of deletion of 17p13, del(17p), varies between 5 and 12% in patients with NDMM [11,108,137,138], and increases as the disease develops, reaching as much as 75% in secondary plasma cell, which is the most aggressive expression of plasma cell dyscrasias [139]. It should be borne in mind that these percentages may vary across different studies depending on the cut-off chosen, although 20% is the most broadly used [140,141,142] and is recommended by the European Myeloma Network (EMN) [143]. This deletion entails the loss of the *TP53* gene, which is a key suppressor gene that organizes multiple functions associated with cell cycle control and DNA damage response [144]. In spite of the infrequent presence of *TP53* mutations in MM, the *TP53* gene is mutated in about half of patients who harbor del(17p), giving rise to its biallelic inactivation [145].

##### Prognosis

Del(17p), although an infrequent genetic abnormality in NDMM, has consistently been associated with shorter survival (Table 1). Nevertheless, there are studies that only found a negative prognostic impact when del(17p) was present in a high proportion, namely between 50 and 60% [146]. Bortezomib-based induction and maintenance regimens seem to improve the prognosis [11,147], according to the HOVON-65/GMMG-HD4 [37] and Total Therapy 3 trials [148] (Table 2). However, response rates and survival are still lower than in patients who do not feature high-risk cytogenetic alterations. Given the results obtained with bortezomib, several studies were carried out with other PIs such as carfilzomib, which yielded better results than those with bortezomib [149], but failed to overcome the poor prognosis associated with del(17p) [149,150]. The trial performed with ixazomib in combination with lenalidomide and dexamethasone gave a better PFS in patients with del(17p) compared with lenalidomide and dexamethasone [151]. The combination of pomalidomide and dexamethasone showed promising results in RRMM patients harboring del(17p), whose PFS was similar to that of patients categorized as being at standard risk. This regimen is thought to be able to overcome the poor prognosis of patients with del(17p), despite the fact that PFS and TTP were less than 1 year (approximately 5 and 8 months, respectively) [40,152].

The introduction of drugs with different mechanisms of action, such as elotuzumab, which is a monoclonal anti-SLAM-F7 antibody, in combination with lenalidomide and dexamethasone, produced better outcomes for patients with del(17p) [41], although the results are difficult to extrapolate since del(17p) was considered positive when at least one cell presented the event, which is at odds with most of the other published studies, which used the threshold of 20% for numerical abnormalities recommended by the EMN [143]. Given this background, there are currently several active trials with elotuzumab in combination with other agents, and even with triplets (elotuzumab + VRD). The anti-CD38 monoclonal antibody isatuximab has also been studied in MM, whereby a phase 2 trial demonstrated better ORR in patients with del(17p) than in those with standard risk (40% vs. 17%) [153].

In essence, del(17p) confers a clear negative impact on survival and has always been considered one of the most important independent prognostic factors. Consequently, del(17p) is one of the high-risk cytogenetic abnormalities included in the R-ISS prognostic index [3,154]. More recently, the bi-allelic inactivation of *TP53* due to the presence of mutations in one allele and deletion in the other has been considered a marker of ultra-high-risk disease in spite of the incorporation of new drugs. In fact, this genetic event is also included in the definition of “double-hit” myeloma along with the combination of 1q amplification and ISS stage 3 [105,155].

##### Therapeutic Implications

The *POL2RA* gene, mapped at 17p, can be lost along with *TP53* in del(17p). This gene encodes an RNA polymerase II enzyme, which can be inhibited by amanitins [156,157]. This drug, conjugated with a targeted antibody against BCMA, has been tested in MM cell lines and mouse models with promising results [158]. Furthermore, the decrease of polymerase activity can be compensated by the overexpression of RBX1. Against this background, combinations of RBX1 inhibitors with amanitins have been proposed to treat patients with metastatic prostate cancer harboring del(17p) [159].

### 2.3. Mutations

Whole-genome and whole-exome sequencing (WGS/WES) by next-generation sequencing of thousands of MM samples has led to the detection of around 60 exonic mutations per patient, with an average 1.6 mutations per Mb [4]. This frequency of mutations is higher than that in acute leukemias, but much lower than that in solid tumors, which tend to have hundreds of mutations [166]. In contrast with other hematological malignancies, there is no universal, unique, and specific mutation in MM, although many recurrently mutated genes have been detected. In fact, about 250 mutated genes have been described in MM [166], about 60 of which are considered driver genes [167]. Most mutations are single nucleotide variants with consequences for the structure of the final protein. These can be present at both the clonal and sub-clonal levels, and evolve as the disease develops.

Various WGS and WES studies have been carried out to characterize the mutational landscape of MM at diagnosis [109] and relapse [168,169]. Some have even compared the landscapes on both occasions by using paired samples [155,170]. Even though there are many mutated genes, only a few of them are mutated in more than 5% of patients [171]: *KRAS* (20–25%), *NRAS* (20–25%), *TP53* (8–15%), *DIS3* (11%), *FAM46C* (11%), *BRAF* (6–15%), *TRAF3* (3–6%), *ROBO1* (2–5%), *EGR1* (4–6%), *SP140* (5–7%), and *FAT3* (4–7%) [172].

#### 2.3.1. Prognosis

Many of the genes that are mutated in MM can be grouped into different pathways whose deregulation may be essential for myeloma cells to survive and proliferate (Figure 2). The pathway most frequently affected by mutations is the mitogen-activated protein kinase (MAPK) pathway, which includes the *KRAS*, *NRAS*, *BRAF*, *EGR1*, and *FGFR3* genes, whose mutations have been detected in almost 50% of MM patients. Although *KRAS* mutations remain neutral, *NRAS* mutations are associated with worse outcomes in patients treated with bortezomib in monotherapy [173], and those affecting *BRAF* are known to have a negative influence on survival [174]. Conversely, mutations of the *EGR1* gene have a favorable effect on outcomes [109]. Mutations affecting the DNA repair pathway, such as those involving the *TP53*, *ATR*, *ATM*, and *ZFHX4* genes, occur in approximately 15% of patients. All these genes, in conjunction with CHEK1/2, participate in this pathway, which is responsible for controlling the apoptotic process in response to DNA damage, among other functions. These mutations have been considered prognostic biomarkers associated with shorter survival [105,109]. *TRAF3*, a component of the NF-kB pathway, which regulates the inflammatory response and cell proliferation, is mutated in about 5% of MM patients. Other components of this pathway, such as *NFKBIA*, *BIRC2/3*, and *CYLD*, are also recurrently mutated, but at frequencies less than 5%. Overall, these mutations account for approximately 20% of the mutations observed in MM. A favorable impact on PFS and OS has been described for *TRAF3* mutations [4,105,175]. *DIS3* and *FAM46C* are two of the genes most frequently mutated after *KRAS* and *NRAS*. Both belong to the RNA processing pathway, which is responsible for regulating gene expression depending on specific environmental conditions. Even though *FAM46C* acts as a tumor-suppressor gene in MM [106,176], no prognostic impact has yet been demonstrated [105]. On the other hand, *DIS3* mutations have recently been described as associated with a deleterious effect on the outcomes of patients treated with intensive therapy [174]. Mutations affecting genes involved in the cell-cycle control pathway, such as *CCND1/2/3*, *CDK4/6*, and *RB1*, are essential for the immortality that characterizes the oncogenic process. Accordingly, mutations in *CCND1*, *CDKN2C*, and *RB1* genes are associated with an unfavorable prognosis [105,109]. Genes participating in the B cell differentiation pathway are also recurrently targeted by mutations. Notably, *SP140* mutations affect about 5% of patients, the gene being implicated in the immune system response mediated by B cell receptors [169]. Another gene prone to mutation is *PRDM1*, whose role in promoting terminal differentiation of B cells is controlled by IRF4 [4]. Strikingly, *IRF4* mutations seem to confer good prognosis [109]. Cereblon participates jointly with IRF4 in the mechanism of action of IMIDs, and its mutations have been associated with shorter PFS and OS in patients treated with lenalidomide-based regimens [177]. Lastly, deregulation of the JAK–STAT pathway by *STAT3* mutations has a negative impact on PFS and OS [178].

Other genes that are affected by mutations but are not included in the aforementioned main pathways include *ROBO1*, *FAT3*, and *ATP13A4*, which are known to play a role in cell migration, adhesion, and regulation of neurons during development [172]. *HUWE1*, mutated in 5% of MM cases, encodes a ubiquitin ligase that blocks MYC-activated genes [179]. Finally, mutations of TGDS gene have been related to a negative prognosis, shortening both OS and PFS [105,179].

#### 2.3.2. Therapeutic Implications

In recent years, our knowledge of the genetic bases of MM has improved considerably, although this has not yet been translated into more precise molecular classifications aimed at providing personalized medicine for treating MM. Current efforts focus on targeted therapy driven by molecular and cytogenetic abnormalities.

Therapeutic agents that inactivate different pathways have demonstrated efficacy in clinical trials at early stages (Table 3). In this context, several clinical trials in various types of cancer, including MM (TAPUR, MATCH, and CAPTURE trials) have begun. Another phase 1/2 clinical trial, MyDRUG (Myeloma-Developing Regimens Using Genomics), has been developed exclusively for RRMM.

CDK mutations, despite having no impact on survival, might be essential to oncogenesis. Two CDK inhibitors, dinaciclib and palbociclib, are effective in RRMM, alone or in combination with bortezomib and dexamethasone, respectively [180,181]. Two case reports of a combination of BRAF and MEK inhibitors have been published [182,183].

TP53 mutations have a clearly unfavorable impact on survival and its poor prognosis remains despite the broad range of treatments available. Consequently, different approaches have been used to increase the activity of the p53 protein [144]. One of them restores the proper protein tertiary structure of mutant p53 by zinc metallochaperone agents [184], PRIMA-1 [185], or APR246 [186], and the other impairs the binding of wild type p53 with its inhibitor, MDM2, by means of nutlin [187], RITA [188], benzodiazepines [189], and spiro-oxindole [190]. Additionally, cyclotherapy pauses the cell cycle and gives wild-type p53 cells an advantage over cells with mutant p53 [191]. Apart from that, *CDK1/6*, *PLK1*, *AURKA*, *NEK2*, and *MYC* genes, among others, are known to be synthetic lethal genes in patients with mutated TP53 [192], which suggests new therapeutic possibilities.

Furthermore, checkpoint inhibitors have been proposed to treat patients with a high rate of mutations, with patients with MAPK mutations having the highest rates [193].

## 3. Future Perspectives

The survival of patients with MM has been significantly extended in the last two decades thanks to the use of triple-drug combinations including PI, IMIDs, and glucocorticoids, and the approval of new agents with proven efficacy against MM. At the same time, we have witnessed a revolution in genomic methodologies that have revealed the enormous genetic complexity, in terms of the number of chromosomal and molecular alterations present in each patient, and of the clonal evolution that these abnormalities undergo in the course of the disease.

Nevertheless, our extensive knowledge of the molecular mechanisms that contribute to the development and progression of MM is yet to have any great impact on therapeutic decisions. In recent years, substantial progress has been made towards enabling more accurate prognostic stratification. However, considerable heterogeneity remains within each cytogenetic risk category, and this makes it challenging to recommend risk-adapted therapeutic solutions, except in the case of the benefit that PI-based regimens provide to high-risk MM patients. On the other hand, it will become increasingly necessary to reformulate new risk stratification models that incorporate not only the prognostic effect of a few cytogenetic abnormalities, but also the weighted impact of each of the numerous genetic lesions that can be detected in every patient with MM [201]. Nor can it be said that the promise of precision medicine will be a reality in the therapeutic approach of MM in the near future, given the difficulty of eradicating genetically highly heterogeneous cell populations whose sub-clonal content evolves over time.

Faced with this scenario, the scientific community will need to direct its efforts in several directions simultaneously. First, it should create platforms and repositories encompassing as much genomic information from standardized NGS panels as possible. Second, it should aim to generate prognostic and predictive biomarkers that allow the currently available therapeutic arsenal to be individualized in such a way as to avoid ineffective drug combinations and thereby minimize unnecessary toxicities. Finally, in parallel, it should promote clinical trials with innovative designs, particularly with respect to the modality of master protocols, in which genetic biomarkers guide the increasing number of therapeutic alternatives.

## Figures and Tables

**Figure 1 cells-10-00336-f001:**
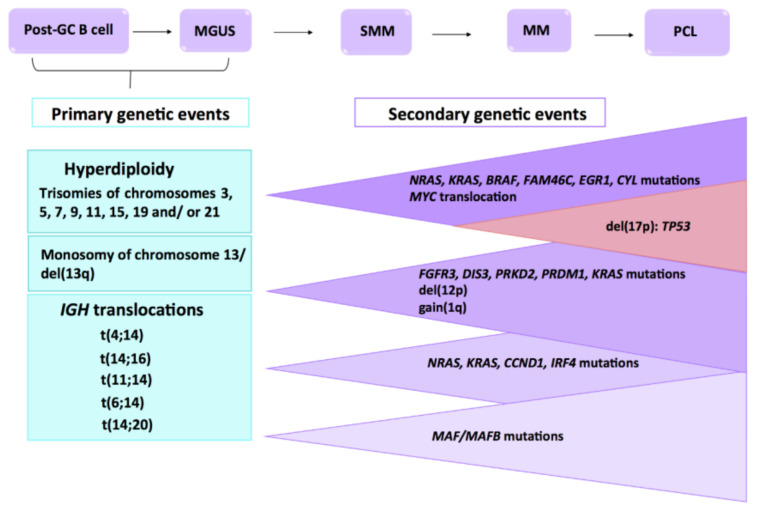
Multistep molecular pathogenesis of multiple myeloma (MM) Primary and secondary genetic events involved in the MM transformation and progression from the precursor disease entities, such as monoclonal gammopathy of undetermined significance (MGUS) and smoldering multiple myeloma (SMM) to MM, and eventually to extramedullary myeloma/plasma cell leukemia (PCL). Adapted from Chesi M et al. and Manier S et al. [4,5].

**Figure 2 cells-10-00336-f002:**
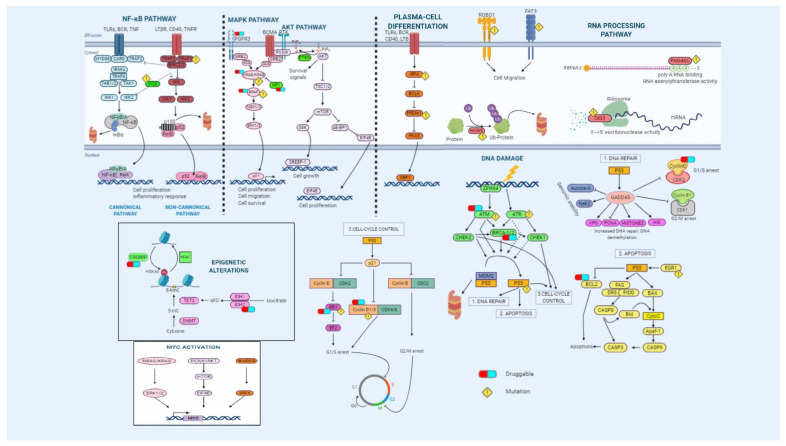
Overview of the main signaling pathways dysregulated in multiple myeloma. Genetic alterations affect essential pathways and biological processes such as NF-κB and MAPK pathways, plasma-cell differentiation, RNA processing, cell cycle, apoptosis, and DNA-damage repair mechanisms. Some druggable targets are indicated.

**Table 1 cells-10-00336-t001:** Prognosis for each cytogenetic abnormality according to the risk stratification models for multiple myeloma.

	mSMART [10]	IFM [160]	IMWG [3]
**t(11;14)**	Standard risk		
**t(4;14)**	Intermediate risk	High risk	High risk
**t(14;16)**	High risk		High risk
**Del(17p)**	High risk	High risk	High risk
**Del(13q) ***	Intermediate risk		
**Gain 1q**	High risk **		

* By metaphase cytogenetic. ** According to https://www.msmart.org/mm-treatment-guidelines. mSMART: Stratification for Myeloma and Risk-Adapted Therapy. IFM: Intergroupe Francophone du Myelome. IMWG: International Myeloma Working Group

**Table 2 cells-10-00336-t002:** Impact of cytogenetic abnormalities on survival outcomes in MM patients according to different treatment regimens.

Trial/Reference	Patients/N	Treatment	Survival Outcomes				
			t(4;14)	del(17p)	t(14;16)	1q gain	t(11;14)
**IFM99-02, 03, 04** [72,108]	NDMM ≤ 65 years/ 1064	IFM-02: VAD+auto-SCTx2 +/- Pam vs Tm+Pam IFM-03: VAD+auto-SCT+allo-SCT IFM-04: VAD+auto-SCTx2 +/- IL6 Inhibitor	- Negative vs. non- t(4;14) - No differences	- Negative vs. non-del(17p) - No differences			- Neutral - No differences
**IFM2005****-01** [33]	NDMM/507	Vd vs. VAD	- Negative vs. non-t(4;14) - Vd improved EFS and OS	- Negative vs. non-del(17p) - No differences			
**TT****2 and TT3** [148]	NDMM/668 NDMM/303	TT2: VAD/CECD+/-T+auto-SCTx2+T+INF TT3:VTD+auto-SCTx2 +VTD+VTD/VRDm		- Negative in TT2 vs. neutral in TT3			
**HOVON65/ GMMG-HD4** [37]	NDMM/399	PAD-auto-SCTx2-Vm vs. VAD-auto-SCTx2-Tm	- Negative vs. non-t(4;14) - No differences	- Negative - Bortezomib arm improved PFS and OS	- Neutral	- Negative - No differences	- Neutral - No differences
**Eloquent-2** [41,161]	RRMM/646	ERd vs. Rd	- Negative - ERd improved PFS	- Neutral - ERd improved PFS			
**Aspire** [161,162]	RRMM/792	KRd vs. Rd	- Negative vs. SR	- Negative vs. SR			
**MM****-003** [40]	RRMM/455	Pd vs. HDd	- Negative vs. SR - Pd improved PFS	- Neutral vs. SR - Pd improved PFS			
**Tourmaline** [151]	RRMM/722	IRd vs. Rd	- No differences	- No differences			
**E****ndeavor** [40,149]	RRMM/929	Kd vs. Vd	- Kd improved PFS	- No differences			
**GIMEMA MM-03-05 + EMN01** [163]	NDMM Ineligible for auto-SCT/474	VMP vs. Rd+Rm	- VMP improved PFS	- No differences	- No differences		
**Myeloma I****X + Myeloma XI** [124,164]	NDMM/1905	IX: CVAD/MP/CTD-auto-SCT-Tm XI: CTD/CRD/CVD-auto-SCT-Rm	- Negative vs. non-t(4;14) - Rm improved PFS	- Negative vs. non-del(17p) - Rm had no impact on PFS	- Negative vs. non-t(14;16) - Rm had no impact on PFS	- Negative vs. non-1q gains - Rm improved PFS	
**Myeloma X** [165]	RRMM/297	Second auto-SCT vs. CFM	- No differences	- Not evaluable			- No differences

Abbreviations: auto-SCT, autologous stem cell transplantation; CECD, cyclophosphamide + etoposide + cisplatin + dexamethasone; CFM, cyclophosphamide; CRD, cyclophosphamide + lenalidomide + dexamethasone; CTD, cyclophosphamide + thalidomide + dexamethasone; CVAD, cyclophosphamide + vincristine + doxorubicin + dexamethasone; CVD, cyclophosphamide + bortezomib + dexamethasone; ERd, elotuzumab + lenalidomide + dexamethasone; HDd, high dose dexamethasone; allo-SCT, allogeneic stem cell transplantation; IRd, ixazomib + lenalidomide + dexamethasone; Kd, carfilzomib + dexamethasone; KRd, carfilzomib + lenalidomide + dexamethasone; PAD, bortezomib + doxorubicin + dexamethasone; Pam, pamidronate maintenance; Pd, pomalidomide + dexamethasone; Rd, lenalidomide + dexamethasone; Rm, lenalidomide maintenance; T, thalidomide; Tm, thalidomide maintenance; VAD, vincristine + doxorubicin + dexamethasone; Vd, bortezomib + dexamethasone; Vm, bortezomib maintenance; VTD, bortezomib + thalidomide + dexamethasone; VRDm, bortezomib + lenalidomide + dexamethasone maintenance. NDMM, new diagnosed multiple myeloma; RRMM, relapsed or refractory multiple myeloma. SR, standard risk; PFS, progression free survival; OS, overall survival.

**Table 3 cells-10-00336-t003:** Targeted therapy in MM.

Gene Mutations/ Chromosomal Abnormalities	Signaling Pathways	Mechanisms of Action	* Targeted Drugs	Clinical Trial (Identifier)/ Reference	Trial Design/Phase
***ATM***	DNA damage repair	PARP inhibitor	Olaparib	NCT02693535	Master protocol (basket)/2
***BRCA1/2***	DNA damage repair	PARP inhibitor	**Veliparib** + Bortezomib	NCT01495351	Traditional design/1
		PARP inhibitor	Olaparib	NCT03297606	Master protocol (basket)/2
		PARP inhibitor	Talazoparib	NCT02693535	Master protocol (basket)/2
		WEE1 inhibitor	Adavosertib	NCT02465060	Master protocol (basket)/2
***BRAF^V600^***	MAPK	BRAF inhibitor	Vemurafenib	NCT01524978/[194]	Master protocol (basket)/2
		MEK inhibitor BRAF inhibitor	**Cobimetinib** + **Vemurafenib**	NCT03297606	Master protocol (basket)/2
		MEK inhibitor AKT inhibitor	**Trametinib + Afuresertib**	NCT01476137/[195]	Traditional design/1–2
		MEK inhibitor BRAF inhibitor	**Trametinib +/- Dabrafenib**	NCT02465060/[196] NCT03091257	Master protocol (basket)/2 Traditional design/1
		BRAF inhibitor MEK inhibitor	**Encorafenib + Binimetinib**	NCT02834364	Traditional design/2
***BRAF*** ***non-V600***	MAPK	ERK1/ERK2 inhibitor	Ulixertinib	NCT02465060	Master protocol (basket)/2
		MEK inhibitor	Selumetinib	NCT01085214/[197]	Traditional design/2
***KRAS***	MAPK	MEK inhibitor BRAF inhibitor	**Trametinib + Afuresertib**	NCT01476137/[195]	Traditional design/1–2
		MEK inhibitor BRAF inhibitor	**Trametinib +/- Dabrafenib**	NCT02465060/[196] NCT03091257	Master protocol (basket)/2 Traditional design/1
		MEK inhibitor	**Cobimetinib** + Pomalidomide + Ixazomib + Dexamethasone	NCT03732703	Master protocol (basket)/1–2
		MEK inhibitor	**Selumetinib** + Panobinostat	[198]	Preclinical study (cell lines)
***NRAS***	MAPK	MEK inhibitor	**Cobimetinib** + Pomalidomide + Ixazomib + Dexamethasone	NCT03732703	Master protocol (basket)/1–2
		MEK inhibitor	Binimetinib	NCT02465060	Master protocol (basket)/2
		MEK inhibitor BRAF inhibitor	**Trametinib +/- Dabrafenib**	NCT02465060/[196] NCT03091257	Master protocol (basket)/2 Traditional design/1
		MEK inhibitor	**Cobimetinib** + Pomalidomide + Ixazomib + Dexamethasone	NCT03732703	Master protocol (basket)/1–2
		MEK inhibitor	**Selumetinib** + Panobinostat	[198]	Preclinical study (cell lines)
***FGFR3***	MAPK	FGFR inhibitor	Erdafitinib	NCT02465060	Master protocol (basket)/2
		FGFR inhibitor	**Erdafitinib** + Ixazomib + Pomalidomide + Dexamethasone	NCT03732703	Master protocol (basket)/1–2
		VEGFR/PDGFR/ CSFR inhibitor	Sunitinib	NCT02693535	Master protocol (basket)/2
		FGFR inhibitor	AZD4547	NCT02465060	Master protocol (basket)/2
***CCND1*** ***CCND3***	Cell cycle	CDK4/6 inhibitor	**Palbociclib** + Bortezomib + Dexamethasone	NCT00555906/[180]	Traditional design/2
		CDK4/6 inhibitor	Palbociclib	NCT02465060	Master protocol (basket)/2
***IDH2***	Epigenetics	IDH2 inhibitor	**Enasidenib** + Ixazomib + Pomalidomide + Dexamethasone	NCT03732703	Master protocol (basket)/1–2
***NF1***	MAPK	MEK inhibitor	Trametinib	NCT02465060	Master protocol (basket)/2
**t(11;14)**	Cell cycle	BCL2 inhibitor	Venetoclax	NCT01794520/[17]	Traditional design/1
		BCL2 inhibitor	**Venetoclax** + Carfilzomib + Dexamethasone	NCT02899052	Traditional design/2
		BCL2 inhibitor	**Venetoclax** + Pomalidomide + Dexamethasone	NCT03567616 NCT03539744	Traditional design/2 Traditional design/3
		BCL2 inhibitor	**Venetoclax** + Bortezomib + Dexamethasone	NCT01794507/[19] NCT02755597/[21]	Traditional design/1 Traditional design/3
		BCL2 inhibitor	**Venetoclax** + Daratumumab + Bortezomib + Dexamethasone	NCT03314181	Traditional design/1–2
		BCL2 inhibitor	Venetoclax	NCT03878524	Master protocol (basket)/1
		BCL2 inhibitor	**Venetoclax** + Ixazomib + Pomalidomide + Dexamethasone	NCT03732703	Master protocol (basket)/1–2
**t(4;14)**		VEGFR/FGFR/ PDGFR inhibitor	Dovitinib	NCT01058434/[47]	Traditional design/2
		BET inhibitor	INCB054329	[77]	Preclinical study (cell lines)
***MYC*** **translocations**	MYC activation	BET inhibitor	**CPI203** + Lenalidomide + Dexamethasone	[199]	Preclinical study (cell lines and patient samples)
		BET inhibitor	OTX015	NCT01713582/[200]	Traditional design/1
		BET inhibitor	Molibresib	NCT01943851	Traditional design/2
**Del13q (*RB* gene)**	Cell cycle	CDK4/6 inhibitor	Palbociclib	NCT02465060	Master protocol (basket)/2

* The targeted drugs when used in combination are indicated in bold.

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
