# Peer review of "Genetic Abnormalities in Multiple Myeloma: Prognostic and Therapeutic Implications"

_cells, 2021, doi:10.3390/cells10020336_

Round 1
Reviewer 1 Report
The authors widely described the genetic abnormalities in multiple myeloma focusing on prognostic and therapeutic aspects.
The paper lacks of an esplicative figure showing the involvement of each abnormalities in the progression of the disease through different stages (MGUS, SMM, MM. In particular, it should be useful to indicate primary and secondary genetic events in each stage.
Moreover, the authors should summarize in a table the mechanisms of resistance to different drugs associated with genetic abnormalities. This aspect is reported in the text but should be highlighted in a table.
Lastly, since the authors mentioned different drugs currently used or in clinical trial, they should add a figure showing the main mechanisms of these drugs
Reviewer 2 Report
This is a well written review on genetics in myeloma covering the major mechanisms of alterations, their prognosis and how treatments can change the unfavourable outcomes. It is split in 3 parts : translocations, copy number alterations and mutations. The references are exhaustive.
Minor revisions :
- I would add one table summarizing the prognosis per se for each abnormalities showing the differences between the Mayo stratification, the IFM and the IMWG in terms of definitions of high, intermediate and standard risk. Moreover, in reference to the recently published paper by J Corre and H Avet Loiseau in Blood 2020, i would discuss why genomic risk is currently changing.
- The threshold for 17p del is 50-60% for the IFM
- Double hit myeloma is not only ISS3 with 1q ampl, it involves also del 17p. Give a better definition
- Table 1 : What is alo ? allogeneic transplantation ?
- present horizontally
Round 2
Reviewer 1 Report
I thank the authors for your complete reply